# Significant Associations between AXIN1 rs1805105, rs12921862, rs370681 Haplotypes and Variant Genotypes of AXIN2 rs2240308 with Risk of Congenital Heart Defects

**DOI:** 10.3390/ijerph17207671

**Published:** 2020-10-21

**Authors:** George Andrei Crauciuc, Mihaela Iancu, Peter Olah, Florin Tripon, Mădălina Anciuc, Liliana Gozar, Rodica Togănel, Claudia Bănescu

**Affiliations:** 1Genetics Laboratory, Center for Advanced Medical and Pharmaceutical Research of George Emil Palade University of Medicine, Pharmacy, Science, and Technology of Targu Mures, 540139 Targu Mures, Romania; andrei.crauciuc@umfst.ro (G.A.C.); florin.tripon@umfst.ro (F.T.); mada_anciuc@yahoo.com (M.A.); claudia.banescu@umfst.ro (C.B.); 2Genetics Department, George Emil Palade University of Medicine, Pharmacy, Science, and Technology of Targu Mures, 540139 Targu Mures, Romania; 3Department of Medical Informatics and Biostatistics, “Iuliu Hatieganu” University of Medicine and Pharmacy Cluj Napoca, 400000 Cluj Napoca, Romania; 4Medical Informatics and Biostatistics Department, George Emil Palade University of Medicine, Pharmacy, Science, and Technology of Targu Mures, 540139 Targu Mures, Romania; 5Pediatrics III Department, George Emil Palade University of Medicine, Pharmacy, Science, and Technology of Targu Mures, 540139 Targu Mures, Romania; liliana.gozar@umfst.ro (L.G.); rodica.toganel@umfst.ro (R.T.)

**Keywords:** congenital heart defects, rs1805105, rs12921862, rs370681, rs2240308

## Abstract

This study aimed to investigate possible associations of the susceptibility to congenital heart defects (CHDs) with *AXIN1* rs1805105, rs12921862 and rs370681 gene variants and haplotypes, and *AXIN2* rs2240308 gene variant. Significant associations were identified for *AXIN1* rs370681 and *AXIN2* rs2240308 variants. *AXIN1* rs370681 variant was significantly associated with decreased odds of CHDs (adjusted OR varying from 0.13 to 0.28 in codominant, dominant and recessive gene models), while the *AXIN2* rs2240308 variant was associated with increased odds of CHD in the dominant model. The haplotype-based generalized linear model regression of *AXIN1* rs1805105, rs12921862 and rs370681 variants revealed that C-C-C and C-C-T haplotypes significantly increased the risk of CHDs (*p* < 0.05). No significant second order epistatic interactions were found between investigated variants (*AXIN1* rs1805105, rs12921862, rs370681, and *AXIN2* rs2240308). Our conclusion is that *AXIN1* rs1805105, rs12921862, and rs370681 (C-C-C and C-C-T) haplotypes and *AXIN2* rs2240308 contribute to CHDs susceptibility.

## 1. Introduction

Congenital heart defects (CHDs) are the most common birth anomaly and occur in approximately 1% of live births [1]. Epidemiological studies revealed that genetic or environmental factors were identified in 25% of cases with CHDs [2]. Recently, one of the largest genetic studies in congenital heart diseases established that 8% of cases had “de novo” autosomal dominant variants, and 2% were due to autosomal recessive inheritance [3]. From a clinical point of view, genetic causes in CHDs are important to be identified for the following reasons: to anticipate the patients’ evolution and their outcome, to assess the risk of recurrence in their offspring and other close relatives, and to evaluate the possible involvement in other extracardiac congenital malformations.

Axis inhibition protein 1 (AXIN1) and Axis inhibition protein 2 (AXIN2) are two components of the β-catenin protein destruction complex (β-CDC). β-CDC had an essential role for the normal function of the Wnt/β-catenin pathway [4]. The mentioned signaling pathway represents one fundamental pathway for embryonic development, cell proliferation regulation, polarity, and tissue homeostasis [5,6]. AXIN1 codified by the *AXIN1* gene, and AXIN2 by the *AXIN2* gene, regulate the activity of the Wnt/β-catenin signaling pathway [4]. Variants of *AXIN1* and *AXIN2* genes, or of the other components of the signaling pathway, may influence the gene activity and also β-catenin mRNA levels [7], therefore, being involved in the development and/or progression of diseases such as cryptorchidism [8], congenital septal heart defects [9], hepatocellular carcinoma [10], colorectal carcinoma [11], and lung and other types of cancers [12]. Previous experimental studies demonstrated that *AXIN1* and *AXIN2* had a crucial role in cell proliferation and heart development [13]. Also, experimental studies such as the study performed by Lei Ji et al. [14] confirmed that several variants of *AXIN1* and *AXIN2* genes were pathogenic variants associated with heart abnormalities [14]. However, based on the latest versions of genome databases (NCBI Homo sapiens Annotation Release 109; Ensembl Release 98), currently several variants are considered to have uncertain significance.

*AXIN1* rs1805105 (c.762 T > C) is a synonymous variant. *AXIN1* rs12921862 (c.878 + 14221G > T) and rs370681 (c.878+3687G > A) are intronic variants of the same gene. *AXIN2* rs2240308 (c.148C > T) is a missense variant (p.Pro50Ser). All of them are currently considered by the mentioned genome databases and by VarSome browser as benign variants. However, the mentioned variants were reported to be associated with high risk, negative prognosis, and various clinical features of different diseases. Li K et al. [15] reported a significant association between two investigated variants of *AXIN1* (rs12921862, rs1805105) and susceptibility for dilated cardiomyopathy [15]. The same variants (rs12921862, rs1805105) are described by Pu Y et al. [16] as being associated with an increased risk for clear cell renal carcinoma [16]. Other studies revealed that *AXIN1* (rs12921862, rs370681) variants were associated with a severe type of bladder cancer [5], and *AXIN2* rs2240308 variant was associated with an increased risk of prostate cancer [17].

However, due to the limited number of studies that aimed to investigate the mentioned variants, the results regarding their clinical significance related to CHD are not clear. According to our knowledge, only two published studies investigated one or two of the mentioned variants in patients with congenital septal heart defects and none of them investigated the *AXIN2* rs2240308 variant in relation to CHD. Taking into consideration their role in oncology, in this field there are numerous published studies that highlight the clinical significance of the mentioned variants.

Considering the current knowledge, this case-control study aimed to investigate if *AXIN1* rs1805105, rs12921862, rs370681; *AXIN2* rs2240308 variants; and *AXIN1* haplotypes contribute to CHDs susceptibility and to estimate the risk conferred by these genes. In addition, we evaluated the *AXIN1* rs1805105, rs12921862, rs370681 and *AXIN2* rs2240308 pairwise interactions.

## 2. Materials and Methods

This case-control study included 214 Caucasian subjects divided into two groups. The control group consisting of 111 healthy children while the patient’s group consisted of 103 CHDs patients from different regions of Romania admitted in the Clinic of Pediatric Cardiology, Emergency Institute for Cardiovascular Diseases and Transplantation Targu Mures, Romania. In the patients group, children with Atrial septal defect (ASD) (51.5%), Ventricular septal defect (VSD) (12%), Tetralogy of Fallot (TOF) (15%), or Double-outlet right ventricle (DORV) (3%) and patients with the combined ASD and VSD (18.5%) were included. The family history was negative in all cases, and the questionnaire completed by the mothers did not highlight prenatal exposure to environmental risk factors related to CHDs (such as maternal medication, alcohol or smoking exposure, parental consanguinity). Patients with a clinical phenotype suggestive for a syndrome (such as Marfan syndrome, Turner syndrome, etc.) or with other congenital malformations were not included in this study. The control group included voluntary non-hospitalized children from different region of Romania, with normal echocardiogram examination, and hospitalized children referred to Pediatrics department for different acute conditions without any congenital structural malformations.

Clinical and demographical data of children included in the study were collected using clinical examinations and medical records (including transthoracic echocardiographic examination) and a questionnaire for the parents or legal guardian of the child.

Ethical approvals were obtained from the Ethical Committee of George Emil Palade University of Medicine, Pharmacy, Science, and Technology of Targu Mures, Romania (47/23.02.2018), and from the Emergency Institute for Cardiovascular Diseases and Transplant Targu Mures. The applied ethical principles were according to the Declaration of Helsinki. The subjects were included in the research activity after a consent, signed by their parents, was obtained.

### 2.1. Genotype Investigation

The DNA was obtained from buccal cells collected with oral swabs (Isohelix, SK-2S, Cell Projects, Kent, UK). For DNA extraction, we used the PureLink Genomic DNA kit (Invitrogen, Carlsbad, CA, USA) and the recommendations of the manufacturer. DNA concentration and absorbance were measured by spectrophotometry (Eppendorf BioSpectrometer, Eppendorf AG, Hamburg, Germany).

For genotyping *AXIN1* rs370681 and *AXIN2* rs2240308, we used Real-Time Polymerase Chain Reaction (PCR) on 7500 Fast Dx system (Applied Biosystem, Foster City, CA, USA) with TaqMan assays (Thermo Fisher Scientific, Waltham, MA, USA), namely C___2221642_20 and C___2577354_1_. Genotyping of *AXIN1* rs1805105 and rs12921862 was performed using polymerase chain reaction-restriction fragment length polymorphism (PCR-RFLP) using specific primers and restriction enzymes, as previously reported [18].

Ensemble genome browser and the Variant Effect Prediction tool of this database were used to annotate and describe the variants.

### 2.2. Data Analysis

Quantitative baseline patients’ characteristics are presented as median with interquartile range (IQR) while qualitative characteristics were represented as absolute and relative frequencies. Mann-Whitney and Chi-square tests were performed for nonparametric data.

Statistical analysis was made in R software v.4.0.0 [19].

Each variant frequency genotype distribution was assessed for departure from Hardy–Weinberg Equilibrium (HWE) using an exact Chi-square test among CHD cases and controls. The exact Chi-square test from “genetics” R package was used [20].

The associations between the studied gene variants and odds of CHD were tested by unconditional logistic regression. According to our knowledge, the age and gender of children were not considered as potential factors for CHD risk in the pediatric population and as a consequence, we did not perform adjustment for age and gender, and for each variant, the odds ratio (OR) and 95% confidence interval (CI) without adjustment for age and gender were estimated. We used four inheritance gene models (codominant, dominant, recessive, and overdominant) to assess the associations between variants and odds of CHD using “SNPassoc” R package [21]. The same package was used to test the potential pairwise interaction between the investigated variants.

Haplotype estimation was performed using R package haplo.stats. The most common haplotype with a significant negative association with the disease risk served as the reference/baseline category. A generalized linear model (GLM) of the haplo.stats package without an interaction term was used.

All statistical tests were two-sided, the threshold of statistical significance being set at α = 0.05. In the case of four inheritance gene models used in binomial logistic regression, the crude *p*-values were adjusted for multiple testing via Benjamini–Hochberg method.

## 3. Results

The ages range from 1 month to 17.5 years in both controls and CHD patients. The median age of CHD patients and control subjects was 6 years (interquartile range [IQR]: 2.0–11.1) and 3 years (interquartile range [IQR]: 0.5–9.0), respectively (*p* = 0.001).

There was no significant difference in frequency distribution by gender between cases and controls (*p* = 0.193); in the patients’ group, 54 (52.42%) were boys and 49 (47.58%) girls, while in controls, 56 (50.45%) were boys and 55 (49.55%) girls.

Concerning prenatal risk factors for congenital heart disease, there was a significant mean difference in maternal age at conception between CHD and control groups (*p* = 0.012); the mean (SD) of maternal age for CHD patients was 26.9 ± 5.6 years and 25.1 ± 4.9 years for control group. The investigated mothers disclaimed the maternal medication, alcohol or smoking exposure during pregnancy. Similarly, type 1 diabetes, lupus and exposure to nonfertility medications during pregnancy were not observed except for five cases with maternal urinary tract infections that were treated with natural drugs.

The investigated variants were successfully genotyped. The genotypes were in Hardy–Weinburg equilibrium (HWE) in both the control and CHD group, except rs1805105 for patients’ group (*p* = 0.0279). Strong or moderate linkage disequilibrium was noticed between *AXIN1* rs1805105 and *AXIN1* rs12921862, and between *AXIN1* rs1805105 and *AXIN1* rs370681, in both control (D’ = 0.66, *p* < 0.01) and CHD patients groups (D’ = 0.41, *p* < 0.001).

The genotypes distributions in CHD and control groups under four inheritance gene models: codominant, dominant, recessive, and overdominant is presented in Table 1. In addition, we explored the difference in genotype distribution in controls and in ASD patient group (with or without VSD). As it may be observed, ASD was the most common CHD found in the present study (72 cases, 69.9%).

No significant differences in genotype frequencies were found for *AXIN1* rs1805105 and rs12921862 (*p* > 0.05), while the genotype frequencies of *AXIN1* rs370681 and *AXIN2* rs2240308 differed significantly in CHD group and controls under the investigated gene models (Table 1). The variant genotypes of *AXIN1* rs370681 were significantly associated with decreased odds of CHDs in the codominant model (adjusted OR = 0.13, 95% CI: 0.05–0.37, *p* = 00002, *p*_BH_ = 0.00004), dominant model (adjusted OR = 0.28, 95% CI: 0.11–0.72, *p* = 0.005, *p*_BH_ = 0.006), and recessive model (adjusted OR = 0.27, 95% CI: 0.14–0.49, *p* = 0.00001, *p*_BH_ = 0.00004). The variant genotypes of *AXIN2* rs2240308 were associated with significant increased odds of CHD in the dominant model after adjusting for maternal age (adjusted OR = 2.01, 95% CI: 1.04–3.90, *p* = 0.036, *p*_BH_ = 0.144). The effect size and direction of association between these two gene variants (*AXIN1* rs370681 and *AXIN2* rs2240308) were similar for ASD susceptibility (Table 1). Moreover, there was a marginally significant unadjusted effect of *AXIN1* rs1805105 variant in the recessive gene model (*p* = 0.048, OR = 0.51, 95% CI: 0.25–1.01) and after adjusting for maternal age group, *AXIN1* rs1805105 gene variant was significantly associated with increased odds of ASD (*p* = 0.041, adjusted OR = 0.48, 95% CI: 0.24–0.99).

In addition, the *AXIN2 rs2240308* gene variant was a significant risk factor for ASD development under the allelic model (OR = 1.60, 95% CI: 1.05–2.44), while *AXIN1 rs12921862* and *AXIN1rs370681* gene variants were protective factors (Table 2).

### 3.1. AXIN1 rs1805105, rs12921862, and rs370681 Haplotypes and Their Associations with CHD Risk

The haplotype analysis of *AXIN1* rs1805105, rs12921862, and rs370681, revealed eight haplotypes. After eliminating one haplotype with an estimated frequency smaller than 0.021, seven haplotypes were used in the haplotype-based association analyses. Haplotypes distribution and their associations with CHDs risk are illustrated in Table 3. The frequency of haplotype C-T-C was lower in CHD cases compared to controls (25.8% versus 33.5%, *p* = 0.026). To identify possible risk haplotypes and quantify the effect of each haplotype, we performed a haplotype-based GLM regression, in which haplotype C-T-C was chosen as reference. The results of model fitting showed that C-C-C and C-C-T haplotypes remained significantly associated with increased CHD risk after adjusting for maternal age (Table 3).

The same haplotypes (C-C-C and C-C-T) had a significant positive association with ASD risk after accounting for maternal age, playing the role of risk factors with an increased risk varying from 4.97 for C-C-C versus C-T-C haplotype and 3.90 for C-C-T versus reference haplotype (Table 4).

### 3.2. Epistatic Pairwise Interactions between AXIN1 rs1805105, rs12921862, rs370681 and AXIN2 rs2240308

Epistatic pairwise interactions between the investigated variants are illustrated in Table 5. No significant second order epistatic interactions were found.

## 4. Discussion

AXIN protein and Wnt/β-catenin canonical mechanisms were described as being involved in cardiovascular diseases; disfunction of AXIN protein and Wnt/β-catenin canonical mechanism may cause congenital heart disease [22,23,24]. Wnt pathway plays an important role in the early stage of heart development [25], and inhibition of the β-catenin gene in the endodermal development stage leads to abnormal cardiac development [26].

Similarities and interactions between *AXIN1* and *AXIN2* genes and between their codified proteins were reported previously [13,27]. Variants of these genes may increase the level of AXIN protein with a normal level of mRNA, suggesting that elevated levels of protein may be explained by the increased stability of AXIN protein [28].

To the best of our knowledge, the present study is the first one that evaluates four variants of the *AXIN* gene (namely *AXIN1* rs1805105, rs12921862, and rs370681 and *AXIN2* rs2240308) at the same time and to investigate their interactions.

Our study showed that *AXIN1* rs1805105 and rs2921862 variants were not associated with susceptibility to CHDs. A significantly negative association was identified for *AXIN1* rs370681 variant, suggesting a potential protective effect of the variant (T) allele. Contrary to our results, Pui Y et al. [18] identified a significant association between variant genotype of the *AXIN1* rs12921862 and ASD susceptibility, with an approximately 3-fold increased risk [18]. Moreover, two of the investigated variants in the *AXIN1* gene, rs12921862 and rs1805105, were reported by Kai L et al. [15] as being associated with an increased risk for dilated cardiomyopathy, in Chinese Han population [15]. The different results may be explained by the multiple types of CHDs (ASD, VSD, TOF, and DORV) included in our study. Other causes may be the small number of cases and different ethnicity of the subjects, with differences being reported in allele frequency, according with gnomAD database (Table 2) across all human populations. Moreover, in our investigated population (pediatric patients and controls), only the frequency of homozygous genotype for *AXIN1* rs12921862 was similar with the data reported in previous studies [15,18]; the frequency of heterozygous genotype was higher in our investigated population, leading to unsignificant results compared to Chinese Population (23.1% in ASD patients and 8.3% in controls reported by Yan Pu et al. [18], 37.1% in dilated cardiomyopathy patients and 8.6% in controls reported by Kai Li et al. [15], compared to 30.1% in our CHD patients and 40.5% in our controls).

Regarding the *AXIN2* rs2240308 variant, our study showed a significant association between the variant genotypes and CHDs risk in the dominant model. According to the latest published data, our study is the first one who analyzed the *AXIN2* rs2240308 variant on CHDs patients. Previous studies evaluated the mentioned variant on patients with hypodontia and demonstrated that the presence of a variant allele was associated with frontal agenesis [29]. Also, the mentioned variant was previously associated with cancer risk, such as colorectal cancer [30], breast cancer [31], and Hirschsprung disease [32]. In the case of CHDs patients, other *AXIN*2 variants [c.28 C > T (p.L10F), c.395 A > G (p.K132R)] were previously reported by Zhu M et al. [33] as pathogenic variants for CHDs.

The haplotype analysis for *AXIN1* variants performed in the current study identified significant associations between CHDs and C-C-C and C-C-T haplotypes, the presence of the mentioned haplotypes being risk factors for CHD development. The difference between the mentioned haplotypes is the latest allele, the allele of *AXIN1* rs1805105 variant. Moreover, our data showed that C-C-C and C-C-T haplotypes had a significant positive association with ASD risk. The OR in the case of the *AXIN1* rs1805105 C allele (OR = 3.49, 95% CI = 1.12–10.87) is higher compared to that for *AXIN1* rs1805105 T allele (OR = 2.88, 95% CI = 1.51–5.51). Separately, in the univariate analysis, *AXIN1* rs1805105 variant was not associated with CHD risk but our results for haplotype analysis suggest that in the case of C-C-C and C-C-T haplotypes, the risk effect is modulated by *AXIN1* rs1805105 variant, with a higher risk for *AXIN1* rs1805105 C allele.

Subsequently, we performed an epistatic pairwise interaction among studied variants to evaluate if the investigated variants interact. Even if the interaction between *AXIN1* and *AXIN2* genes and between their codified proteins was described previously [13], our results did not find epistasis interactions between the investigated variants. No data have been available from other studies until now for variant interactions for *AXIN1* rs1805105, rs12921862, and rs370681 and *AXIN2* rs2240308.

Regarding the HWE, we observed that *AXIN1* rs1805105 was not HWE in the CHDs group. However, the allele frequencies found in our study are similar to those reported in the literature for different populations [15], and according to Ensembl genome browser, all of our investigated variants had similar allele frequencies to those reported by the genome browser.

Briefly, our results showed that the presence of the variant genotypes of *AXIN2* rs2240308 and the presence of the C-C-C and C-C-T haplotypes of *AXIN1* (rs1805105, rs12921862, rs370681) are associated with a high risk of CHD development.

Concerning prenatal risk factors for congenital heart disease, an increase in maternal age at conception was observed (the mothers of CHD patients 26.9 ± 5.6 years and 25.1 ± 4.9 years for control group). Our observation is in line with that of Fung A et al. [34].

Maternal urinary tract infections were recorded in five cases, and no association was observed with CHD in the study group similarly to the results previously reported by Fung A et al. [34].

Although a family history of CHD, presence of associated congenital anomalies, maternal smoking, and alcohol consumption during pregnancy are reported to be associated with increased odds of CHD according to Fung A et al. [34], our study could not highlight these associations as long as we had cases with no history of CHD and cases with a clinical phenotype suggestive for a syndrome. Moreover, we did not include patients with other congenital malformations.

The strength of our study consists in the simultaneous analysis of four variants of *AXIN* genes. According to our knowledge, there are no other studies that performed a haplotype analysis and epistatic pairwise interaction between the investigated variants. Also, our study is the first one to evaluate the *AXIN2* rs2240308 on CHDs patients. Moreover, a significant association between the variant genotypes of *AXIN2* rs2240308 and CHD risk was noticed. On the other hand, our study has some limitations. One of these is the lack of measurement of the Axin protein levels in our subjects. The small number of subjects represents another limitation of our study, therefore, further large-scale studies are required to validate our results. Another limitation was the lack of whole exome sequencing analysis and data regarding socio-economic, educational status and body mass index of the parents.

## 5. Conclusions

The present study is an exploratory research that reveals significant positive associations between susceptibility to congenital heart defects and *AXIN1* rs1805105, rs12921862 and rs370681 (C-C-C and C-C-T) haplotypes and *AXIN2* rs2240308 variant.

## Figures and Tables

**Table 1 ijerph-17-07671-t001:** Genotypes distribution and their associations with odds of CHD/ASD.

Variants	Gene Models	Genotypes	Controls (%), *n* = 111	CHD Cases (%), *n* = 103	OR Crude ^a^ (95% CI)	OR Adjusted ^b^ (95% CI)	ASD Cases (%), *n* = 72	OR Crude ^a^ (95% CI)	OR Adjusted ^b^ (95% CI)
***AXIN1*** **rs1805105**	Codominant	TT	14 (12.6)	15 (14.6)	1.00	1.00	11 (15.3)	1.00	1.00
		TC	59 (53.2)	63 (61.2)	1.00 (0.44–2.24)	1.16 (0.50–2.68)	46 (63.9)	0.99 (0.41–2.39)	1.15 (0.46–2.90)
		CC	38 (34.2)	25 (24.3)	0.61 (0.25–1.49)	0.66 (0.27–1.64)	15 (20.8)	0.50 (0.19–1.35)	0.54 (0.19–1.51)
	Dominant	TT	14 (12.6)	15 (14.6)	1.00	1.00	11 (15.3)	1.00	1.00
		TC + CC	97 (87.4)	88 (58.4)	0.85 (0.39–1.85)	0.95 (0.42–2.13)	61 (84.7)	0.80 (0.34–1.88)	0.90 (0.37–2.19)
	Recessive	TT + TC	73 (65.8)	78 (75.7)	1.00	1.00	57 (79.2)	1.00	1.00
		CC	38 (34.2)	25 (24.3)	0.62 (0.0.34–1.12)	0.59 (0.0.32–1.09)	15 (20.8)	0.51 (0.25–1.01)	**0.48 (0.24–0.99)**
	Overdominant	TT + CC	52 (46.8)	40 (38.8)	1.00	1.00	26 (36.1)	1.00	1.00
		TC	59 (53.2)	63 (61.2)	1.39 (0.81–2.39)	1.53 (0.87–2.70)	46 (63.9)	1.56 (0.85–2.87)	1.73 (0.92–3.28)
***AXIN1*** **rs12921862**	Codominant	CC	64 (57.7)	68 (66.0)	1.00	1.00	47 (65.3)	1.00	1.00
		CA	45 (40.5)	31 (30.1)	0.65 (0.37–1.15)	0.62 (0.35–1.12)	24(33.3)	0.73 (0.39–1.35)	0.68 (0.35–1.29)
		AA	2 (1.8)	4 (3.9)	1.88 (0.33–10.63)	1.68 (0.29–9.83)	1 (1.4)	0.68 (0.06–7.73)	0.63 (0.05–7.95)
	Dominant	CC	64 (57.7)	68 (66.0)	1.00	1.00	47 (65.3)	1.00	1.00
		CA + AA	47 (42.3)	35 (34.0)	0.70 (0.40–1.22)	0.67 (0.38–1.19)	25 (34.7)	0.72 (0.39–1.34)	0.67 (0.36–1.28)
	Recessive	CC + CA	119 (98.2)	99 (96.1)	1.00	1.00	71 (98.6)	1.00	1.00
		AA	2 (1.8)	4 (3.9)	2.20 (0.39–12.29)	2.00 (0.35–11.54)	1 (1.4)	0.77 (0.07–8.62)	0.73 (0.06–9.06)
	Overdominant	CC + AA	66 (59.5)	72 (69.9)	1.00	1.00	48 (66.7)	1.00	1.00
		CA	45 (40.5)	31 (30.1)	0.63 (0.36–1.11)	0.61 (0.34–1.09)	24 (33.3)	0.73 (0.39–1.36)	0.68(0.36–1.30)
***AXIN1*** **rs370681**	Codominant	CC	7 (6.3)	20 (19.4)	1.00	1.00	16 (22.2)	1.00	1.00
		CT	49 (44.1)	61 (59.2)	0.44 (0.17–1.11)	0.43 (0.16–1.14)	44 (61.1)	0.39 (0.15–1.04)	0.37 (0.13–1.03)
		TT	55 (49.5)	22 (21.4)	**0.14 (0.05–0.38)**	**0.13 (0.05–0.37)**	12 (16.7)	**0.10 (0.03–0.28)**	**0.09 (0.03–0.27)**
	Dominant	CC	7 (6.3)	20 (19.4)	1.00	1.00	16 (22.2)	1.00	1.00
		CT + TT	104 (93.7)	83 (80.6)	**0.28 (0.11–0.69)**	**0.28 (0.11–0.72)**	57 (77.8)	**0.24 (0.09–0.61)**	**0.22 (0.08–0.60)**
	Recessive	CC + CT	56 (50.5)	81 (78.6)	1.00	1.00	60 (83.3)	1.00	1.00
		TT	55 (49.5)	22 (21.4)	**0.28 (0.15–0.50)**	**0.27 (0.14–0.49)**	12 (16.7)	**0.20 (0.10–0.42)**	**0.19 (0.09–0.41)**
	Overdominant	CC + TT	62 (55.9)	42 (40.8)	1.00	1.00	28 (38.9)	1.00	1.00
		CT	49 (44.1)	61 (59.2)	**1.84 (1.07** **–3.16)**	**1.90 (1.08** **–3.32)**	44 (61.1)	**1.99 (1.09** **–3.64)**	**2.02 (1.08** **–3.77)**
***AXIN2*** **rs2240308**	Codominant	GG	34(30.6)	18 (17.5)	1.00	1.00	11 (15.3)	1.00	1.00
		GA	55 (49.5)	60 (58.3)	**2.06 (1.05-4.06)**	**2.00 (1.00–4.00)**	41 (56.9)	**2.30 (1.04–5.08)**	2.22 (0.99–5.01)
		AA	22 (19.8)	25 (24.3)	2.15 (0.96–4.82)	2.04 (0.90–4.65)	20 (27.8)	**2.81 (1.13–6.98)**	**2.69 (1.06–6.85)**
	Dominant	GG	34 (30.6)	18 (17.5)	1.00	1.00	11 (15.3)	1.00	1.00
		GA + AA	77 (69.4)	85 (82.5)	**2.09 (1.09** **–3.99)**	**2.01 (1.04** **–3.90)**	61 (84.7)	**2.45 (1.15–5.23)**	**2.36 (1.08–5.14)**
	Recessive	GG + GA	89 (80.2)	78 (75.7)	1.00	1.00	52 (72.2)	1.00	1.00
		AA	22 (19.8)	25 (24.3)	1.30 (0.68–2.48)	1.26 (0.65–2.45)	20 (27.8)	1.56 (0.78–3.12)	1.54 (0.75–3.15)
	Overdominant	GG + AA	56 (50.5)	43 (41.7)	1.00	1.00	31(43.1)	1.00	1.00
		GA	55 (49.5)	60 (58.3)	1.42 (0.83–2.44)	1.41 (0.81–2.45)	41 (56.9)	1.35 (0.74–2.45)	1.32 (0.71–2.46)

Note. ^a^—unadjusted regression analysis; ^b^—adjusted for maternal age treated as categorical variable (defined by quartiles) in the logistic regression model; range of maternal age values (years) for quartile groups: first quartile group: (18; 21); second quartile group: (22; 25); third quartile group: (26; 29); fourth quartile group: (30; 42). Significant results were achieved when *p*-values < 0.05 and the corresponding OR and CI values are highlighted with bold.

**Table 2 ijerph-17-07671-t002:** Allele frequency evaluation and their associations with odds of CHD/ASD.

Variants	Variant Alleles	Overall Variant Allele Frequency/European Allele Frequency/East Asian Allele Frequency (% gnomAD v2.1.1) ^a^	Variant Allele Frequency in Control Group (95% CI)	Variant Allele Frequency in CHD Group (95% CI)	OR Crude ^b^ (95% CI)	Variant Allele Frequency in ASD Group (95% CI)	OR Crude ^b^ (95% CI)
***AXIN1* rs1805105**	C allele	61.5/64.2/29.9	60.8 (54.1–67.3)	54.9(47.8–61.8)	0.78 (0.53–1.15)	52.8 (44.3–61.1)	0.72 (0.47–1.10)
***AXIN1* rs12921862**	A allele	17.4/18.3/20.4	22.1 (16.8–28.1)	18.9(13.8–24.9)	0.83 (0.52–1.32)	18.1 (12.1–25.3)	**0.36 (0.21** **–** **0.62)**
***AXIN1* rs370681**	T allele	48.2/54.3/30.6	71.6 (65.2–77.5)	50.9(43.9–57.9)	**0.41 (0.28** **–** **0.61)**	47.2 (38.9–55.7)	**0.36 (0.23** **–** **0.55)**
***AXIN2* rs2240308**	A allele	46.6/52.5/32.6	44.6 (37.9–51.4)	53.4 (46.3–60.4)	1.42 (0.97–2.08)	56.3 (47.7–64.5)	**1.60 (1.05** **–** **2.44)**

Note. ^a^ allele frequency in worldwide populations and European (non-Finnish) population from Genome Aggregation Database (gnomAD; https://gnomad.broadinstitute.org); 95% CI = 95% confidence interval; ^b^—unadjusted odds ratio in the allelic model; Significant results were achieved when *p*-values < 0.05 and the corresponding OR and CI values are highlighted with bold.

**Table 3 ijerph-17-07671-t003:** Haplotype results for association of *AXIN1* gene with CHD risk.

Haplotypes rs12921862 rs370681 rs1805105	Hap-Freq in Control Group	Hap-Freq in CHD Patients	Hap-Score ^a^	*p* ^b^	OR Crude ^c^, 95% CI (Lower-Upper Limit)	OR Adjusted ^d^, 95% CI (Lower-Upper Limit)
Global test of association for additive model with covariate: Statistics = 21.22, df = 6, Global *p*-value = 0.0016						
*C-T-C*	0.339	0.258	−2.226	0.0260	1.00 (Reference)	1.00 (Reference)
*C-T-T*	0.207	0.134	−1.958	0.0502	1.21 (0.54–2.73)	1.18 (0.52–2.71)
*A-T-C*	0.142	0.118	−1.807	0.0708	0.96 (0.41–2.26)	0.98 (0.41–2.35)
*A-T-T*	0.028	0.000	−1.293	0.1962	0.69 (0.12–4.04)	0.73 (0.12–4.67)
*A-C-C*	0.051	0.052	1.365	0.1722	2.63 (0.87–7.91)	2.59 (0.84–7.97)
*C-C-C*	0.076	0.126	2.247	**0.0246**	**3.49 (1.12–10.87)**	**3.84 (1.16–12.67)**
*C-C-T*	0.157	0.293	3.448	**0.0006**	**2.88 (1.51–5.51)**	**2.96 (1.52–5.77)**
*A-C-T*	0.000	0.020	NA	NA	NA	NA

Note. Haplotypes estimated from the three variants are ordered according to the Score test statistics; ^a^ haplotypes frequencies inferred by haplo.stats; haplotype score for the haplotype; ^b^
*p*-values of the corresponding Score test; ^c^ effect sizes of each haplotype estimated from haplotype-based GLM regression without covariates; ^d^ effect sizes of each haplotype adjusted for maternal age treated as categorical variable (defined by quartiles); NA = not available because of their low relative frequency (<0.021); *p*-values smaller than 0.05 and the corresponding OR and CI values are highlighted with bold.

**Table 4 ijerph-17-07671-t004:** Haplotype results for association of *AXIN1* gene with ASD risk.

Haplotypes rs12921862 rs370681 rs1805105	Hap-Freq in Control Group	Hap-Freq in ASD Patients	Hap-Score ^a^	*p* ^b^	OR Crude ^c^, 95% CI (Lower-Upper Limit)	OR Adjusted ^d^, 95% CI (Lower-Upper Limit)
Global test of association for additive model with covariate: Statistics = 25.80, df = 6, Global *p*-value = 0.0002						
*C-T-C*	0.339	0.236	−2.641	0.0083	1.00 (Reference)	1.00 (Reference)
*C-T-T*	0.207	0.121	−1.977	0.0481	1.38 (0.53–3.61)	1.32 (0.48–3.64)
*A-T-C*	0.142	0.115	−1.759	0.0785	1.09 (0.39–3.00)	0.98 (0.33–2.90)
*A-T-T*	0.028	0.000	−1.152	0.2489	0.67 (0.07–6.43)	0.46 (0.04–4.96)
*A-C-C*	0.051	0.052	1.431	0.1525	2.89 (0.73–11.38)	3.05 (0.76–12.31)
*C-C-C*	0.076	0.125	2.278	**0.0227**	**4.64 (1.20–17.96)**	**4.97 (1.16–21.22)**
*C-C-T*	0.157	0.337	4.062	**0.00004**	**3.96 (1.86–8.40)**	**3.90 (1.79–8.49)**
*A-C-T*	0.000	0.014	NA	NA	NA	NA

Note. Haplotypes estimated from the three variants are ordered according to the Score test statistics; ^a^ haplotypes frequencies inferred by haplo.stats; haplotype score for the haplotype; ^b^
*p*-values of the corresponding Score test; ^c^ effect sizes of each haplotype estimated from haplotype-based GLM regression without covariates; ^d^ effect sizes of each haplotype adjusted for maternal age treated as categorical variable (defined by quartiles); NA = not available because of their low relative frequency (<0.021); *p*-values smaller than 0.05 and the corresponding OR and CI values are highlighted with bold.

**Table 5 ijerph-17-07671-t005:** Epistatic pairwise interactions between *AXIN1* and *AXIN2* variants.

Gene Polymorphisms	Genetic Model	*AXIN1* rs1805105	*AXIN1* rs12921862	*AXIN1* rs370681	*AXIN2* rs2240308
*AXIN1* rs1805105	Codominant	0.281	0.144	0.669	0.278
	Dominant	0.678	0.048	0.897	0.189
	Recessive	0.111	0.223	0.526	0.358
	Overdominant	0.238	0.561	0.263	0.452
*AXIN1* rs12921862	Codominant	0.314	0.217	0.666	0.435
	Dominant	0.893	0.210	0.570	0.522
	Recessive	0.241	0.359	0.181	0.892
	Overdominant	0.237	0.111	0.445	0.941
*AXIN1* rs370681	Codominant	0.544	0.253	<0.001	0.888
	Dominant	0.619	0.278	0.003	0.915
	Recessive	0.280	0.229	<0.001	0.425
	Overdominant	0.307	0.106	0.027	0.612
*AXIN2* rs2240308	Codominant	0.330	0.376	0.078	0.080
	Dominant	0.729	0.380	0.031	0.024
	Recessive	0.402	0.492	0.347	0.434
	Overdominant	0.258	0.255	0.225	0.203

Note. The *p*-value of epistatic pairwise interactions represented in the upper part of the matrix was obtained using the log-likelihood ratio test (LRT). The *p*-value from the diagonal of the matrix represent unadjusted (crude) effect on CHD of all investigated variants, obtained from LRT. Also, the *p*-value from the lower part of the matrix was obtained using LRT comparing the likelihood of the model for variants of two genes and the best model containing one gene variant.

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
