# Peer review of "Significant Associations between AXIN1 rs1805105, rs12921862, rs370681 Haplotypes and Variant Genotypes of AXIN2 rs2240308 with Risk of Congenital Heart Defects"

_ijerph, 2020, doi:10.3390/ijerph17207671_

Round 1

Reviewer 1 Report

The authors have correctly answered all the requirements. the article, in its present format, is ready to be published

Reviewer 2 Report

The manuscript by Crauciuc and co-authors described the prevalence and associations of AXIN1 and AXIN2 polymorphic variants in patients with congenital heart disorders. The papers includes a detailed statistical analysis and demonstrated the association of several ACIN1 and AXIN2 variant with congenital heart diseases. 

The paper is sientifically sound and contains an important information which is worth to be published. However, it contains a major limitation which does not allow to present it in a a hight rating Q2 journal. It contains insufficient number of the control subjects (100) which does not allow to make a deep comparision of the study and the control group. Taking into account the low number of cases (103) the control group should be extended far above 111 cases. Additionally, the brief description of the control group (how it was collected anywhere the blood samples came from) should be included.

 The data presentation regarding the genotypes has several lints to discussed, for example the terms codominant, overdominant for the combination of the genotypes are not unto date and the additional analysis for the allele frequencies (even if it is not significant) should be performed in such studies. The tables are recommenced to include the GnomAD data on genotypes and allele frequencies along with the control group studies.

 Overall, the manuscript even being scientifically merit, has to be improved mainly interns of a control group (at leat 500 cases), after which it can be resubmitted.

Reviewer 3 Report

Moderate Modification

The authors here presented a case-control study to investigate associations of SNPs present in AXIN1 and AXIN2 with CHD. It seems that the results are very promising, though it lacks some clinical information on factors influencing CHD. This study aimed to investigate possible associations of the susceptibility to congenital heart defects (CHDs) with AXIN1 rs1805105, rs12921862, rs370681 gene variants and haplotypes, and AXIN2 rs2240308 gene variant. Significant associations were identified for AXIN1 rs370681 and AXIN2 rs2240308 variants. AXIN1 rs370681 variant was significantly associated with decreased odds of CHDs (adjusted OR varying from 0.13 to 0.28 in codominant, dominant and recessive gene models), while the AXIN2 rs2240308 variant was associated with increased odds of CHD in the dominant model. This can help a lot for this field and attract many readers. After careful revision, I believe this manuscript meet part of our Journal’s high demaind. If editors consider to publish it faster, I recommend it only need some English words modification.

Reviewer 4 Report

This is an interesting exploratory study that investigated the association between congenital heart disease (CHD) and AXIN1 gene variants and haplotypes, and AXIN2 gene variants in a case-control study that included 103 CHD patients. An interesting approach of this study is the haplotype analysis which demonstrated different odds ratios when the same AXIN variants were assessed. Some minor comments should be adjusted/explained in more detail. 

  1. The authors did not perform adjustment for age and gender of the odds ratios between the studied gene variants and CHD. This should be explained in more detail in the method section.
  2. This study identified a negative association for the AXIN1 rs370681 variant and CHD, while other studies have identified a significant association between variant genotype of the AXIN1 rs12921862 and ASD susceptibility, with an approximately 3-fold increased risk. These differences in results  should be more clearly explained. 

  3. In the haplotype analysis a haplotype-based GLM regression was performed, in which haplotype C-T-C was chosen as reference. Why was this particular C-T-C reference choses as reference? This should be explained.
  4. The Discussion section is rather long and should be more succinct because in its current form it could lead to an overanalysis of the results. 

Round 2

Reviewer 2 Report

Some of the issue have been addressed and the manuscript can reaccepted in the present form.

This manuscript is a resubmission of an earlier submission. The following is a list of the peer review reports and author responses from that submission.

Round 1

Reviewer 1 Report

I have reviewed the study 'Significant associations between AXIN rs1805105, rs12921862, rs370684 haplotypes and variant genotypes of AXIN2 rs2240308 with Risk of congenital heart defects' submitted by Georgre Andrei Crauciuc et al. 

The idea to identify the genetic reasons for congenital heart defects (CHDs) is good, but not really novel. However the design of the experiments is done in a poor way. Today whole exome sequencing (WES) are the standard for These studies. There are many further genes which could be the reasons. To screen AXIN1 and AXIN2 for SNPs is not acceptable. In Addition, all SNPs have a high minor allele frequency (https://gnomad.broadinstitute.org/variant/16-396264-A-G?dataset=gnomad_r2_1) and are definitly above the prevalence of CHD in the healthy Control population.

Therefore the only Suggestion is to reject the manuscript.  

Reviewer 2 Report

The authors designed a case-control study to investigate associations of SNPs present in AXIN1 and AXIN2 with CHD. Despite results are promising, the lack of clinical information on factors influencing CHD jeopardise the result extrapolation. 

Major Points: 

a) Please describe the population from which samples were obtained. Some SNPs show great frequency variation between different populations, specifically the rs1805105 (https://www.ensembl.org/Homo_sapiens/Variation/Population?db=core;r=16:345764-346764;v=rs1805105;vdb=variation;vf=22711755)

b) The composite of Congenital Hearth Disease could be influenced by the most prevalent disease (ASD). Please, show, at least, the assotiation of rs370681, rs2240308 and the haplotypes C-C-C and C-C-T with Atrial Septal Defect (with and without VSD). 

c) Is there any information about other factors influencing CHD? Smoking status of mother, alcohol consumption, BMI? Did you collect data on medication, insulin-dependent diabetes, lupus? Is the case group free of other genetic changes that could affect heart (Marfan, Tuner..)? Some factors influencing CHD could be obtained from here (https://www.ahajournals.org/doi/10.1161/JAHA.113.000064). 

Minor Points: 

Line 49: Rephrase “AXIN1 and AXIN2 codified by the AXIN1 gene, respectively, by the AXIN2 gene”.

Line 87: How you check the exposure to environmental risk factors?